# Learning Abstract World Models with a Group-Structured Latent Space

## Abstract

Learning meaningful abstract models of Markov Decision Processes (MDPs) is crucial for improving generalization from limited data. In this work, we show how geometric priors can be imposed on the low-dimensional representation manifold of a learned transition model. We incorporate known symmetric structures via appropriate choices of the latent space and the associated group actions, which encode prior knowledge about invariances in the environment. In addition, our framework allows the embedding of additional unstructured information alongside these symmetries. We show experimentally that this leads to better predictions of the latent transition model than fully unstructured approaches, as well as better learning on downstream RL tasks, in environments with rotational and translational features, including in first-person views of 3D environments. Additionally, our experiments show that this leads to simpler and more disentangled representations.

## 1 Introduction

In recent years, abstract world models have emerged as an important foundation for tackling complex reinforcement learning (RL) problems. World model learning can capture meaningful representations by embedding complex, high-dimensional data into lower-dimensional abstract spaces (Ha and Schmidhuber, 2018; Francois-Lavet et al., 2019; Hafner et al., 2019; Schrittwieser et al., 2020; Kipf et al., 2019; van der Pol et al., 2020a). While the goal of world model learning is improved sample efficiency, state-of-the-art methods (Ye et al., 2021; Hafner et al., 2023) still demand a large number of training samples to achieve superhuman performance.

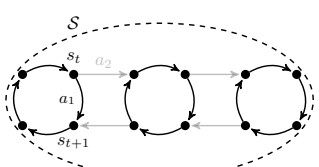

Figure 1: Illustration of an MDP dynamics with some symmetries that we want to use as a prior knowledge in the abstract space geometry.

The agent's performance depends on how the representation space models the environment's underlying dynamics. The representation should be minimal yet sufficient to ensure generalization to unseen scenarios. Prior knowledge can also be leveraged. For instance, it is possible to enforce equivariances to improve generalization in RL (van der Pol et al., 2020b; Wang et al., 2022c; Park et al., 2022).

In decision-making settings, geometric priors allow agents to represent states and actions in a way that respects the underlying symmetry of the dynamics—for instance, an agent that rotates in place eventually returns to its original state. Such priors can enable faster learning and better generalization when the environment exhibits known geometric structure. Consider the simple MDP shown in Figure 1, which contains both symmetric and non-symmetric features. Repeated applications of action $a_1$ bring the agent back to the same state. We propose encoding such symmetries by choosing a latent space and corresponding group action that naturally reflect this structure and the rest is learned (e.g. how much an agent rotates with a given action as well as unstructured features). Our approach employs a coordinate system tailored to the symmetry group, and performs a change of coordinates to recover a Euclidean representation, as illustrated in Figure 2.

The paper is organized as follows: Section 2 provides the background and introduces notations. Section 3 provides the methodology for integrating prior knowledge in the learning of abstract representations. Section 4 reports experiments on different settings including first-person view games in 3D environments. The experiments show the superior performance when generalizing to unseen

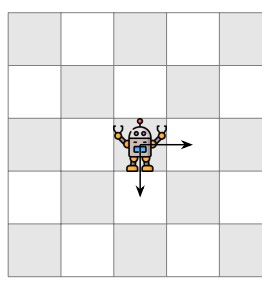 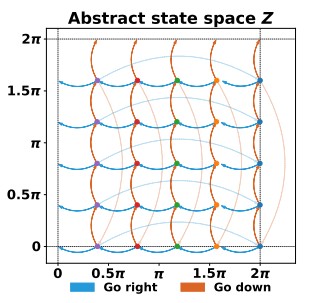 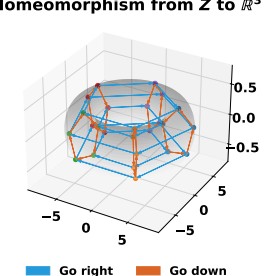

Figure 2: Illustration of the approach. (*Left*) A grid-world with periodic boundary condition where the agent will appear on the opposite side when it attempts to move outside the grid. (*Middle*) The abstract state space $\mathcal{Z}$ is modeled as a product space $\mathbb{R}/2\pi\mathbb{Z} \times \mathbb{R}/2\pi\mathbb{Z}$. (*Right*) Elements of $\mathcal{Z}$ can be mapped to $\mathbb{R}^3$ using a homeomorphism $(x, y) \mapsto ((\alpha + \beta\cos(x))\cos(y), (\alpha + \beta\cos(y))\cos(x), \beta\sin(y))$ for $\alpha, \beta \in \mathbb{R}_+$ specifying the major and minor radius of a torus.

scenarios and in the low-data regime. Section 5 provides further discussion on the related work. Section 6 provides a conclusion that highlights and discusses the main contributions.

## 2 BACKGROUND

### 2.1 GROUP THEORY

This section introduces several essential definitions from group theory and abstract spaces.

**Groups** A group G is defined by a pair $(G, *)$, consisting of a set G and a binary operator $* : G \times G \to G$, such that: (i) there is an *identity element* $e \in G$ satisfying $e * g = g * e = g$ for any $g \in G$, (ii) the operator $*$ is *associative*, i.e., $(g_1 * g_2) * g_3 = g_1 * (g_2 * g_3)$ for any $g_1, g_2, g_3 \in G$, and (iii) every element has an *inverse*, meaning that for any $g \in G$, there exists $h \in G$ such that $g * h = h * g = e$. An example of a group is the *cyclic group* $(\mathbb{Z}/n\mathbb{Z}, +)$, where each element is an integer in $\{0, \ldots, n-1\}$, and for every $u, v \in \mathbb{Z}/n\mathbb{Z}$, we have $u + v := (u + v) \mod n$.

**Group actions.** Groups can also be used to define transformations on another set. In the context of reinforcement learning, applying an action moves the agent from the current state $s_t \in \mathcal{S}$ to the next state $s_{t+1} \in \mathcal{S}$. In this setting, the transition dynamics T defines a *group action* acting on the state space $\mathcal{S}$, which we use as prior knowledge. Formally, a group action $\cdot : G \times \mathcal{S} \to \mathcal{S}$ is a binary operation that satisfies two axioms: $e \cdot s = s$ and $(g * g) \cdot s = g \cdot (g \cdot s)$ for any $g \in G$ and $s \in \mathcal{S}$.

An *orbit* of a state $s \in \mathcal{S}$ under the group action is a set of all states reachable from $s$ via elements of G, i.e., $\text{Orb}(s) := \{g \cdot s \mid g \in G\}$. For example, given a state $s$, all the 90-degree rotations of that state form an orbit of $s$ under the group of 90-degree rotations. The group action G can partition the state space $\mathcal{S}$ into multiple orbits. In Figure 1, the action $a_1$ consistently generates an orbit of 4 states, a pattern that remains predictable across different initial states.

For a toy MDP with a single orbit generated by a single action (illustrated by a single directed cycle in Figure 1), we may represent the state space as $\mathcal{S} := \{z \mid z \in \mathbb{C}, |z| = 1\}$, a set of unit complex numbers. The group action on $\mathcal{S}$ is defined as $g \cdot s = \exp\left(\frac{2\pi i}{n}\right) \times s$, where $g$ is the $n^{\text{th}}$ root of unity. Composing multiple actions generates the set of $n^{\text{th}}$ roots of unity, which forms $\mathbb{Z}/n\mathbb{Z}$ under complex multiplication. Alternatively, when $\mathcal{S}$ is a vector space $\mathbb{R}^n$, the group action on $\mathcal{S}$ is performed via invertible matrices from SO$(n)$ (Quessard et al., 2020).

**Equivalence classes and Quotient spaces.** An *equivalence relation* groups together elements of a set that share a property. If two elements are considered equivalent under the relation, they belong to the same *equivalence class*. An equivalence relation $\sim$ must satisfy three simple rules: (i) *Reflexivity*: every element is equivalent to itself, $s \sim s$, (ii) *Symmetry*: if $s_x \sim s_y$, then $s_y \sim s_x$, and (iii) *Transitivity*: if $s_x \sim s_y$ and $s_y \sim s_z$, then $s_x \sim s_z$.

A *quotient space* arises naturally when we identify equivalent elements within a set. For example, consider the real numbers $\mathbb{R}$ with an equivalence relation defined by translation by multiples of a constant $k \in \mathbb{R}$. In this case, two real numbers $x$ and $y$ are equivalent if their difference is a multiple of $k$, i.e., $x \sim y$ if and only if $x - y \in k\mathbb{Z}$. The quotient space $\mathbb{R}/k\mathbb{Z}$ then consists of equivalence classes of real numbers, where each class contains all real numbers that differ by a multiple of $k$. Intuitively, this quotient space "wraps" the real line $\mathbb{R}$ into a circle of circumference $k$, since adding multiples of $k$ to any point on the line brings you back to an equivalent point on the circle.

## 2.2 MARKOV DECISION PROCESSES

A *Markov Decision Process (MDP)* $\mathcal{M}$ is defined by a $(\mathcal{S}, \mathcal{A}, R, T, \gamma)$-tuple, which includes: (i) a *state space* $\mathcal{S} \subseteq \mathbb{R}^n$ that can be either discrete or continuous, (ii) an *action space* $\mathcal{A}$ that can be either discrete or continuous, (iii) a *reward function* $R : \mathcal{S} \times \mathcal{A} \to \mathbb{R}$, which assigns scalar rewards to the agent's actions at any state, (iv) a *transition dynamics* $T : \mathcal{S} \times \mathcal{A} \to \mathcal{S}$, which captures the transition dynamics of the MDP, and (v) a *discount factor* $\gamma$, which determines the importance of future rewards. In most cases, the agent does not have access to the reward structure $R$ and the transition dynamics $T$ and must rely on interactions with the MDP to approximate solutions.

## 2.3 WORLD MODELS

The main focus of this work is on world models, a framework that aims to approximate an MDP's underlying reward function $R$ and transition dynamics $T$ through interaction. In contrast to existing algorithms that learn world models through pixel reconstruction (Ha and Schmidhuber, 2018; Hafner et al., 2023), our approach is based on a *self-supervised representation learning method* (Francois-Lavet et al., 2019; Gelada et al., 2019; Kipf et al., 2019; van der Pol et al., 2020a; Hansen et al., 2024) that operates in an *abstract representation space* $\mathcal{Z}$.

**Self-Supervised World Models** Self-Supervised World Models usually consist of (i) a learnable encoding $\varphi : \mathcal{S} \to \mathcal{Z}$, (ii) a learnable transition function $\tau : \mathcal{Z} \times \mathcal{A} \to \mathcal{Z}$ and (iii) a learnable reward function $r : \mathcal{Z} \times \mathcal{A} \to \mathbb{R}$. $\varphi$ projects states into a low-dimensional *abstract state space* $\mathcal{Z} \subseteq \mathbb{R}^d$, with $d \ll n$. $\tau$ models the transition dynamics $T$ in the abstract state space. $r$ approximates the reward function $R$. These mappings are parameterized by deep neural networks, with the parameters collectively denoted as $\boldsymbol{\theta} := (\theta_{\text{enc}}, \theta_{\text{trans}}, \theta_{\text{rew}})$. In our approach, $\boldsymbol{\theta}$ are learned jointly by optimizing for a contrastive representation learning objective.

Assuming a fixed exploration policy $\pi_{\text{explore}}$, an agent interacts with the MDP according to $\pi_{\text{explore}}$ and collects experience tuples $(s_t, a_t, r_t, s_{t+1})$ into a replay buffer $\mathcal{D}$.

The states in the sampled tuple $(s_t, a_t, r_t, s_{t+1})$ are mapped into abstract latent states:

$$z_t := \varphi(s_t; \theta_{\text{enc}}); \quad z_{t+1} := \varphi(s_{t+1}; \theta_{\text{enc}}) \tag{1}$$

The next latent state $z_{t+1}$ is modeled as a *translation* from the current state $z_t$ given the action $a_t$:

$$\hat{z}_{t+1} := \tau(z_t, a_t; \theta_{\text{trans}}) \tag{2}$$

The exact functional form of the transition model $\tau(z_t, a_t)$ plays a crucial role in this work as we will show later in Section 3. The agent aims to learn a model with sufficient predictive power of the future, meaning that $z_{t+1} \approx \hat{z}_{t+1}$. This objective is expressed as a predictive loss that encourages alignment between the model's prediction and the ground truth:

$$\mathcal{L}_{\text{trans}}(\theta_{\text{enc}}, \theta_{\text{trans}}) := \mathbb{E}\left[d(\hat{z}_{t+1}, z_{t+1})\right], \tag{3}$$

where $d : \mathcal{Z} \times \mathcal{Z} \to \mathbb{R}_+$ is a measure of similarity in the abstract state space, which can be the $\ell_1$ or $\ell_2$ distance for simplicity.

Optimizing solely for the predictive loss $\mathcal{L}_{\text{trans}}$ would lead to *latent collapse*, where the learned encoder $\varphi$ maps all states to a single point in $\mathcal{Z}$. One approach to prevent collapse is to optimize an entropy loss (Francois-Lavet et al., 2019; Wang and Isola, 2022) that encourages the encoder $\varphi$ to preserve information in the abstract state space $\mathcal{Z}$:

$$\mathcal{L}_{\text{entropy}}(\theta_{\text{enc}}, \theta_{\text{trans}}) := \mathbb{E}\left[\exp\left(-C \cdot d(z_x, z_y)\right)\right], \tag{4}$$

where $z_x \neq z_y$ are random abstract states sampled from $\mathcal{Z}$. The hyperparameter $C$ controls the regularization strength.

The combination of the losses in Equations 3 and 4 is functionally similar to the InfoNCE loss (van den Oord et al., 2019):

$$\mathcal{L}_{\text{InfoNCE}}(\theta_{\text{enc}}, \theta_{\text{trans}}) := \mathbb{E} \left[ -\log \frac{\exp\left(-d(\hat{z}_{t+1}, z_{t+1})/t\right)}{\exp\left(-d(\hat{z}_{t+1}, z_{t+1})/t\right) + \sum_{z_{t+1}^-} \exp\left(-d(\hat{z}_{t+1}, z_{t+1}^-)/t\right)} \right], \tag{5}$$

where the expectation is taken over random transitions in the replay buffer $\mathcal{D}$, $t$ is the temperature parameter, and $z_{t+1}^-$ denotes the negative ground-truth for the latent prediction (Eq. 2) of a given transition $(z_t, a_t, z_{t+1})$. Here, the numerator enforced the fitting of the internal transition function and the denominator ensures sufficient entropy. The InfoNCE was developed in the context of contrastive learning methods. It has also been used in the context of world models (Wang et al., 2022d).

The reward function fits the conditional expectation $\mathbb{E}[R \mid z_t, a_t]$ with the L2 loss:

$$\mathcal{L}_{\text{rew}}(\theta_{\text{rew}}) := \mathbb{E} \left[ \|\mathbf{r}(z_t, a_t) - r_t\|_2^2 \right] \tag{6}$$

## 3 GEOMETRIC PRIORS IN ABSTRACT WORLD MODELS

### 3.1 WORLD MODELING

We use a standard approach to world modeling described in Section 2. Additionally, we regularize the volume of the abstract state space $\mathcal{Z}$ to be small using a hinge loss:

$$\mathcal{L}_{\text{vol}}(\theta_{\text{enc}}, \theta_{\text{trans}}) := \mathbb{E} \left[ \max\left(\|z_{t+1} - z_t\|_2 - w, 0\right) \right], \tag{7}$$

where $w$ is a threshold hyperparameter. This regularization prevents the vector norm $\|z_t\|$ from growing large, which facilitates the visualization and interpretability of the abstract state space $\mathcal{Z}$.

Putting everything together, the world model algorithm used in this work jointly optimizes

$$\mathcal{L}_{\text{abstract}}(\boldsymbol{\theta}) := \mathcal{L}_{\text{InfoNCE}}(\theta_{\text{enc}}, \theta_{\text{trans}}) + \mathcal{L}_{\text{rew}}(\theta_{\text{rew}}) + \mathcal{L}_{\text{vol}}(\theta_{\text{enc}}, \theta_{\text{trans}}) \tag{8}$$

which is minimized using any stochastic gradient descent algorithms.

Overall, the learnable mappings $\varphi$, $\tau$ and r together define an abstract MDP $\widetilde{\mathcal{M}}$ which is a $(\mathcal{Z}, \mathcal{A}, \tau, \mathbf{r}, \gamma)$-tuple. When optimizing the loss in Equation 8, we recover an abstract dynamics that is meaningful. Given the current state $s_t \in \mathcal{S}$, an action $a_t \in \mathcal{A}$, and the next state $s_{t+1} = \mathrm{T}(s_t, a_t)$ as defined by $\mathcal{M}$, the following relation holds:

$$\tau(\varphi(s_t), a_t) = \varphi(s_{t+1}). \tag{9}$$

In addition, the rewards are accurately approximated by $r(z_t, a_t)$.

### 3.2 GEOMETRIC PRIORS IN MDP REPRESENTATIONS

Let us consider again the toy MDP introduced in Figure 1. The group action G acting on $\mathcal{S}$ is assumed to be a cyclic group $\mathbb{Z}/n\mathbb{Z}$. As a first form of geometric priors, we propose to model the abstract state space $\mathcal{Z}$ as the quotient space $\mathbb{R}/k\mathbb{Z}$. In this case, the learnable encoder $\varphi : \mathcal{S} \to \mathcal{Z}$ maps the states of the original MDP $\mathcal{M}$ to elements of equivalence classes $[z] \in \mathcal{Z}$, where $[z] := \{z + k \cdot t \mid t \in \mathbb{Z}\}$.

**Geometric Priors for Transition Models.** In our framework, the transition model $\tau : \mathcal{Z} \times \mathcal{A} \to \mathcal{Z}$ maps to the same abstract space $\mathcal{Z}$. The group action G acting on $\mathcal{S}$ is modeled as *an additive group action on $\mathcal{Z}$*, denoted by $\oplus$. Concretely, for a given abstract state $z_t \in \mathcal{Z}$ and action $a \in \mathcal{A}$, we define the interaction with the learned group action $\Delta(z_t, a)$ to predict the next abstract state $z_{t+1}$ as:

$$\hat{z}_{t+1} = \tau(z_t, a; \theta_{\text{trans}}) := z_t \oplus \Delta(z_t, a; \theta_{\text{trans}}). \tag{10}$$

The exact form of $\oplus$ depends on the algebraic structure of $\mathcal{Z}$. When $\mathcal{Z}$ is the canonical vector space $\mathbb{R}^d$, the operator $\oplus$ corresponds to standard vector addition, representing a translational group action.

In contrast, when $\mathcal{Z}$ is the quotient space $\mathbb{R}/k\mathbb{Z}$, the operator can be implemented via modular arithmetic: for $x, y \in \mathcal{Z}$, we define $x \oplus y := (x + y) \bmod k$.

This additive group action serves as a geometric prior and enables models to learn simpler representations of the underlying dynamics. For a one-step transition, the next abstract state $z_{t+1}$ is assumed to be close to $z_t$. With this assumption, small, incremental changes in the state space are reflected smoothly in the abstract space, ensuring local continuity of the data manifold.

These geometric priors enhance our world model by structuring the abstract state space $\mathcal{Z}$ to reflect the symmetries and regularities of the environment. These improvements come without altering the training objectives or network architectures, making the approach both effective and efficient. In earlier work geometric priors are built directly into equivariant network architectures through weight-tying (e.g. van der Pol et al. (2020b), Park et al. (2022), Wang et al. (2022a)), enhancing sample efficiency. However, the use of equivariant networks comes with additional computational overhead (Satorras et al., 2021; Kaba et al., 2023; Luo et al., 2024). In contrast, we capture symmetries in the MDP through an abstract representation space with additional structure.

### 3.3 JOINT SYMMETRIC AND NON-SYMMETRIC REPRESENTATIONS

Learning a representation that combines both symmetric and non-symmetric features can be challenging. When these features are entangled in the abstract state space, it becomes difficult to correctly apply geometric priors and group actions to the appropriate set of features, while simultaneously identifying which features should remain unaffected by these actions.

To address this challenge, we promote *disentanglement* (Higgins et al., 2018) within the abstract space $\mathcal{Z}$ by regularizing sparsity on the transition vector $\Delta(z, a; \theta_{\text{trans}})$. Specifically, we constrain the features of $\Delta(z, a; \theta_{\text{trans}})$ that should remain invariant with respect to the action $a$ to be zero. This objective is achieved using the following loss function:

$$\mathcal{L}_{\text{disentanglement}}(\theta_{\text{enc}}, \theta_{\text{trans}}) = |\Delta(z, a; \theta_{\text{trans}})^{\sigma(a)}| \tag{11}$$

Here, $\sigma : \mathcal{A} \to \mathcal{I} \subseteq \mathcal{P}(\{1, \ldots, d\})$ is a mapping that specifies the coordinates of the $d$-dimensional abstract state space $\mathcal{Z}$ that remain unaffected by action $a$ and $\Delta(\cdot)^{\sigma(a)}$ denotes a subspace whose coordinates are given by $\sigma(a)$. The mapping $\sigma$ satisfies $\bigcap_{a \in \mathcal{A}} \sigma(a) = \emptyset$, ensuring that each action affects only its respective latent subspace. Additionally, we modify $\mathcal{L}_{\text{InfoNCE}}$ (Eq. 5) so that negative samples $z_{t+1}^-$ are drawn from other transitions involving the same action $a_t$. In other words, we compute $\mathcal{L}_{\text{InfoNCE}}$ by averaging over conditional expectations with respect to actions $a \in \mathcal{A}$.

## 4 EXPERIMENTS

We first consider a simple case: rotational symmetry alone. Next, we evaluate on the Torus MDP that involves multiple symmetric group actions. We then evaluate on VizDoom to demonstrate that our method scales to complex environments with high-dimensional inputs. In all experiments, our approach (i) leads to an interpretable abstract representation, (ii) improves generalization to unseen transitions within the MDP and (iii) improves performance on downstream RL tasks.

### 4.1 IMPLEMENTATION DETAILS

The abstract model learning follows the algorithm outlined in Section B.1. We collect a finite set of tuples with a random policy $\pi_{\text{explore}}$ that selects actions uniformly from $\mathcal{A}$. For most experiments, the encoder $\varphi$ and the transition model $\tau$ are multi-layer perceptrons (MLPs). For the experiments on Vizdoom (Section 4.3), a convolutional neural network (CNN) followed by MLPs are used.

### 4.2 ABSTRACT REPRESENTATIONS FOR ROTATIONAL SYMMETRY

**Representing the** $\mathrm{SO}(2)$ **group** The first experiment involves a simple MDP, named `Passage`, with $n = 7$ states, where the agent can move either left or right, wrapping around to the opposite end upon reaching a boundary. This MDP is represented by a directed $n$-cycle. The state space $\mathcal{S}$ is a collection of one-hot vectors $\{0, 1\}^n$ representing the agent's current position. In this experiment, a single feature is sufficient to capture the abstract representation, as shown in Figure 3.

The group action G in `Passage` is an $n$-order cyclic group $\mathbb{Z}/n\mathbb{Z}$. In fact, any cyclic group is a finite subgroup of the special orthogonal group SO(2), which is isomorphic to the quotient group $\mathbb{R}/k\mathbb{Z}$. Thus, SO(2) $\simeq \mathbb{R}/k\mathbb{Z}$ can be viewed as the limiting group of $\mathbb{Z}/n\mathbb{Z}$ as $n$ increases. These geometric priors indicate that our method can naturally extend to represent continuous group actions.

**Combined symmetry groups** As a natural extension, we consider the `Torus` MDP (see Figure 2), where the transition dynamics imply that the group action is a product of two cyclic groups, G := $\mathbb{Z}/n\mathbb{Z} \times \mathbb{Z}/n\mathbb{Z}$. Each state $s \in \{0,1\}^{2n}$ is a concatenate of two one-hot vectors, one represents the agent's current row and the other the current column. In this MDP, the state space has the topology of a torus. We incorporate this geometric prior by modeling $\mathcal{Z} := \mathbb{R}/k\mathbb{Z} \times \mathbb{R}/k\mathbb{Z}$ with $k := 2\pi$.

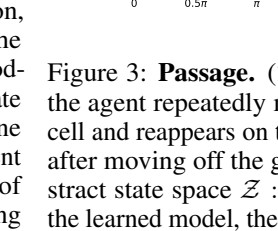

Figure 3: **Passage.** (*Top*) In this MDP, the agent repeatedly moves to the right cell and reappears on the first empty cell after moving off the grid. (*Bottom*) Abstract state space $\mathcal{Z} := \mathbb{R}/2\pi\,\mathbb{Z}$. From the learned model, the next abstract state equals to the previous one plus $2\pi/7$.

As shown in Figure 4, the learned abstract representations accurately capture the transition dynamics T. Additionally, these representations can be mapped to a standard vector space $\mathbb{R}^3$ via a homeomorphism, demonstrating that our method can recover the MDP's structure using a compact latent space of just two dimensions. This result highlights the efficiency of our approach: it preserves the underlying symmetries and significantly reduces the dimensionality of the representation, which is key for scaling to more complex environments.

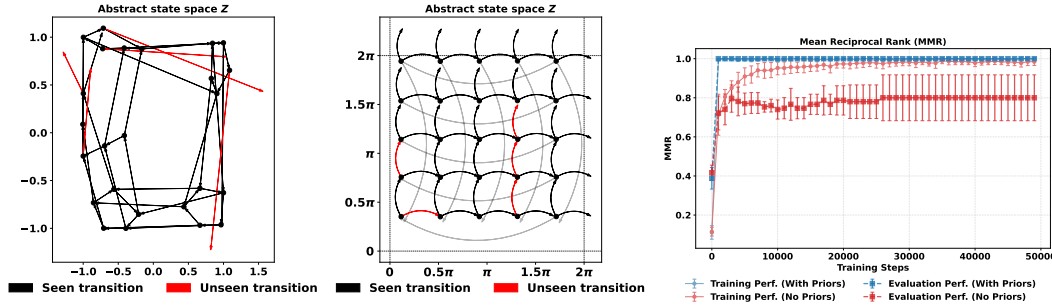

Figure 4: Generalization on unseen transitions predicted by the model (in red) for the `Torus` MDP. In this setting, $10\%$ of the possible state-action pairs are disabled during training. *(Left)* Without geometric priors, the abstract model fails to predict the unseen transitions. *(Middle)* With geometric priors, the abstract model accurately predicts the unseen transitions. *(Right)* Quantitative comparison between the two approaches, measured by the mean reciprocal rank.

### 4.3 COMBINING STRUCTURED AND UNSTRUCTURED FEATURES

In a first-person view, symmetry groups can provide succinct representations. We look at how group transformations (rotations, translations, reflections, etc.) preserve certain structures in the observer's reference frame. The combination of translation ($T^2$ in 2D or $T^3$ in 3D) and rotation (SO(2) or SO(3)) forms the Euclidean group E(2) or E(3), which represents the full set of rigid body motions (moving and rotating) that preserve the observer's perspective. In these experiments, we do not enforce a particular structure on the translation symmetry group and treat it as an unstructured feature.

**Top-down view** The first environment that we consider is a top-down view in the MiniGrid environments (Chevalier-Boisvert et al., 2024). The agent's action space consists of a "move forward" and a "turn right 90°" action, with a constraint that its position must be within an $n \times n$-grid world. The state space $\mathcal{S}$ is a collection of concatenated one-hot vectors $\{0,1\}^{2n+4}$ that jointly encodes the agent's current $(x, y)$-coordinate and orientation {North, East, South, West}. Figure 5 shows how information about rotation and translation can be disentangled in different feature dimensions.

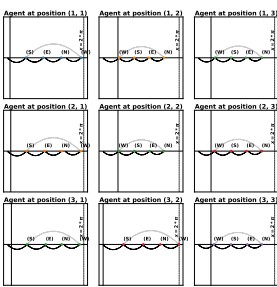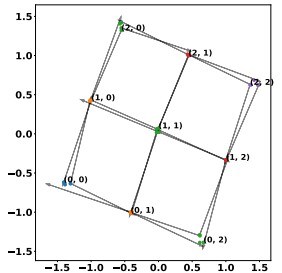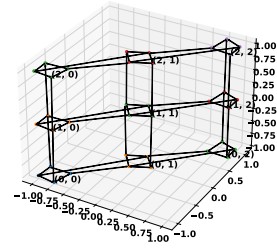

Figure 5: **MiniGrid** ($3 \times 3$). (*Left*) First latent variable $z^{(1)} \in \mathbb{R}/2\pi\,\mathbb{Z}$ that encodes the agent's orientation. (*Center*) The subspace $(z^{(2)}, z^{(3)}) \in \mathbb{R}^2$ which encodes the agent's position in the grid. In this setting, the overall proposed abstract state space $\mathcal{Z}$ is modeled as a product space $\mathbb{R}/2\pi\,\mathbb{Z} \times \mathbb{R}^2$. (*Right*) Abstract state space $\mathcal{Z} \subseteq \mathbb{R}^3$ when modeling without geometric priors.

**First-person view from environments with high-dimension inputs** In this section, we analyze our method and the learned representations in a high-dimensional environment, VizDoom (Kempka et al., 2016). Here, the state space $\mathcal{S}$ is a set of RGB frames $\in [0, 255]^{64 \times 64 \times 3}$ capturing the first-person perspectives. The first-person rotations in VizDoom can be assumed to be continuous or, at the very least, represented by a cyclic group $\mathbb{Z}/n\mathbb{Z}$, where $n$ is large. For instance, in VizDoom, without any modifications, the agent needs to rotate approximately 100 times to complete a 360° turn. In our setup, we fix the rotation angle to $\delta = 36$. A custom map and scenario are used for this experiment (cf. Appendix B.3). Our static dataset consists of 100,000 transitions, divided into a train and validation set. The dataset was generated by a random policy uniformly sampling actions from the set {"forward", "nothing", "turn left", "turn right"}. A variation of the InfoNCE loss is used in the VizDoom experiments; see the experimental details in Appendix B.3.

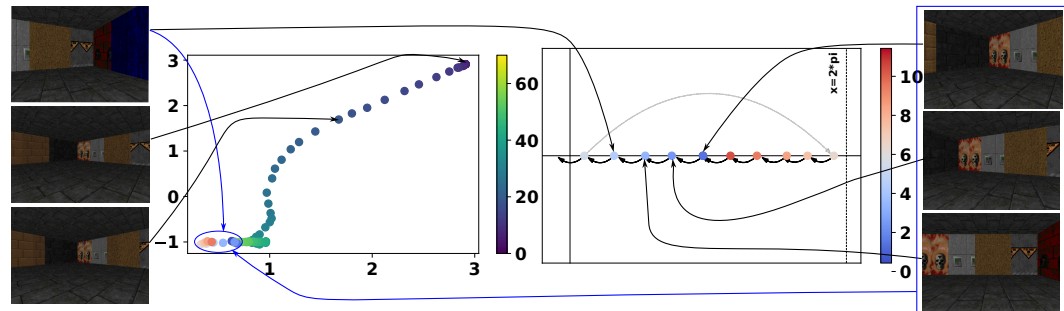

Figure 6: Visualization of the effect of geometric priors on high-dimensional input. The figure shows the mapping of high-dimensional data into a low-dimensional space. (*Left*) Latent subspace $(z^{(2)}, z^{(3)})$ that encodes spatial information. (*Right*) Latent subspace $z^{(1)}$ that encodes orientation.

In Figure 6, the agent moves straight for 30 steps, then remains idle for 30 steps to stop its momentum, and finally performs 10 "turn right" actions. The agent recovers the cyclic structure of the environment in the rotation space. Additionally, the agent stacks all states at the same point in the spatial space during its rotation, indicating that the action-conditioned regularization loss effectively shapes the latent space as intended. Compared to abstract world models without geometric priors, we are able to map high-dimensional input into a very low-dimensional space (only 3 dimensions for VizDoom).

### 4.4 QUANTITATIVE RESULTS ON GENERALIZATION

The model's ability to learn from a finite set of training data can be assessed by its accuracy on previously unseen data points drawn from the same underlying distribution. This evaluation quantifies how well the learned features capture the environment's structure without overfitting.

Two agents are compared: one with and one without a geometric prior by using Hits at k (H@k) and Mean Reciprocal Rank (MRR), two metrics used in earlier work on world modeling Kipf et al. (2019); Park et al. (2022). These metrics are provided in Appendix C. Various training set sizes are

tested to analyze performance in both low- and high-data regimes. Figure 4 shows that priors enable the agent to accurately predict unseen transitions, whereas the agent without priors overfits on seen transitions. Similarly, Table 1 shows that the agent with prior knowledge significantly outperforms both the agent without it and the one with greater representation power (e.g., higher latent dimension) across H@1, H@5, and MRR metrics, across all tested environments. We also compare against the approach of Quessard et al. (2020). While it achieves competitive performance on Torus, its performance degrades significantly on the VizDoom environments, suggesting that our method can handle both structured and unstructured features. Figure 7 highlights that the prior reduces overfitting for 3D first-person environments (similar figures for the Torus and Minigrid environments are in Appendix D).

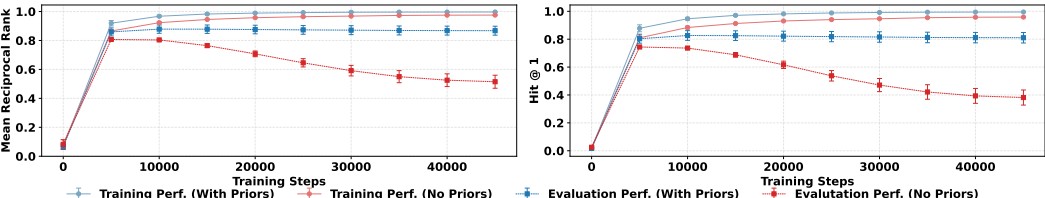

Figure 7: Generalization performance on VizDoom using 10,000 transitions for the train set (10% of the original dataset) and 20,000 transitions for the test set (20% of the original dataset). (*Left*) MRR computed on the training and test sets. (*Right*) H@1 computed on the training and test sets.

Table 1: Ranking results on the validation set (10% of valid transitions for Torus and MiniGrid. 20% for VizDoom). Each metric is multiplied by a factor of 100.

| Environment | Model | H@1 (↑) | H@5 (↑) | MRR (↑) |
|---|---|---|---|---|
| MiniGrid $5 \times 5$ | **AWM + Geometric Priors** | 85.55 (± 14.31) | 97.77 (± 2.72) | 91.05 (± 9.21) |
| | AWM (same latent dimensionality) | 13.33 (± 5.65) | 70.00 (± 13.42) | 37.65 (± 6.26) |
| | PRAE (van der Pol et al., 2020a) | 24.44 (± 10.88) | 77.77 (± 7.85) | 24.44 (± 10.88) |
| | Rotation Matrix (Quessard et al., 2020) | 83.33 (± 0) | 98.15 (± 2.62) | 90 (± 1.2) |
| Torus $5 \times 5$ | **AWM + Geometric Priors** | 96.00 (± 8.00) | 100.00 (± 0.00) | 98.00 (± 4.00) |
| | AWM (same latent dimensionality) | 56.00 (± 8.00) | 100.00 (± 0.00) | 70.40 (± 6.69) |
| | PRAE (van der Pol et al., 2020a) | 12.00 (± 9.79) | 100.00 (± 0.00) | 42.53 (± 6.38) |
| | Rotation Matrix (Quessard et al., 2020) | 100 (± 0.00) | 100 (± 0.00) | 100 (± 0.00) |
| VizDoom | **AWM + Geometric Priors** | 81.04 (± 3.75) | 93.72 (± 2.35) | 86.77 (± 3.09) |
| | AWM (same latent dimensionality) | 59.26 (± 5.02) | 79.09 (± 2.39) | 68.56 (± 3.81) |
| | PRAE (van der Pol et al., 2020a) | 42.42 (± 20.72) | 71.74 (± 12.47) | 55.68 (± 18.71) |
| | Rotation Matrix (Quessard et al., 2020) | 17.58 (± 18.42) | 27.17 (± 14.90) | 23.67 (± 16.48) |

### 4.5 EVALUATION ON DOWNSTREAM REINFORCEMENT LEARNING TASKS

We further evaluate the generalization capability of our approach by measuring how well a reinforcement learning (RL) agent performs when trained with a limited amount of real-world interaction data. In this setting, the agent gets a positive reward when reaching a designated goal in the environment. At each time step, the agent receives a reward of $-1$, and the episode terminates once the goal is reached. Details of the experimental setup are in Appendix F.

We compare three agents: (i) a baseline using Double Q-learning (DDQN) (van Hasselt et al., 2015) alone, (ii) an agent combining DDQN with a learned abstract world model that incorporates a geometric prior, and (iii) an agent combining DDQN with the abstract world model but without the geometric prior. All three agents are trained on the same fixed dataset of transitions, collected using a random policy. The abstract world models are frozen during the DDQN training phase. In the model-based setting, the Q-function is trained on encodings of the raw states. Additionally, the learned transition model is used to generate synthetic transitions to augment the training data. The performance of the agents can be seen in Figure 8.

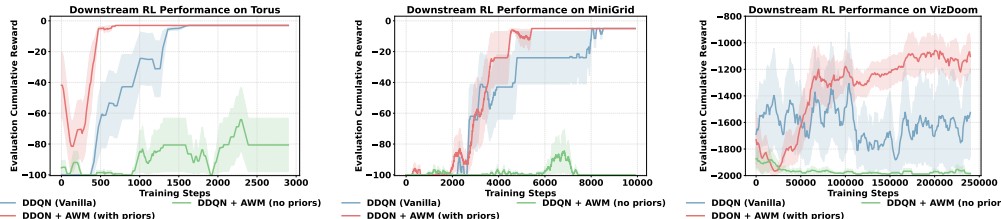

Figure 8: Cumulative rewards averaged over 5 seeds and the corresponding standard error. Each curve shows a running average of the return computed over 100 training steps. The shaded areas depict the standard errors.

## 5 RELATED WORK

**Abstract World Models** Abstract (or latent) world models aim to learn a simplified dynamics of the world by ignoring irrelevant information. Most approaches achieve this by mapping high-dimensional inputs (such as images) into a more compact representation space that should contain the key features of the environment. Various methods have been proposed to construct this abstract space. For instance, Ha and Schmidhuber (2018) and Hafner et al. (2024) introduce approaches based on variational autoencoders (VAEs), where the latent space is learned through probabilistic inference. Other approaches use contrastive learning techniques that bypass input reconstruction (Francois-Lavet et al., 2019; Gelada et al., 2019; Kipf et al., 2019; van der Pol et al., 2020a; Hansen et al., 2024; Park et al., 2022). Several works have enforced structure within the latent space, for instance van der Pol et al. (2020a). Rezaei-Shoshtari et al. (2022) learn MDP homomorphisms in tandem with the policy.

**Geometric Priors** Earlier works have shown that equivariant RL algorithms can have better sample efficiency (van der Pol et al., 2020b; Mondal et al., 2020; Simm et al., 2020; Wang et al., 2022b; van der Pol et al., 2021; Wang et al., 2022a; Zhu et al., 2022; Chen and Zhang, 2023). In these works, the exact group structure of the MDP is assumed to be known. In contrast, our approach assumes only a cyclic group structure without requiring the precise number of elements in the group. Quessard et al. (2020) proposed a similar approach to ours. However, the key differences lie in how rotations are represented and how disentanglement is achieved. In their method, rotations are represented using rotational matrices rather than complex numbers on the unit circle, and disentanglement is not learned through a contrastive objective. Other work enforces group structure in the latent MDP by placing symmetry constraints on the latent transition model (Park et al., 2022). While Quessard et al. (2020) and Park et al. (2022) enforce the use of fully symmetric features, our method allows for the combination of both symmetric and non-symmetric features. This flexibility enables our method to scale to more complex environments, such as first-person games like VizDoom. Finzi et al. (2021) introduces "Residual Pathway Priors" as a mechanism to imbue models with soft inductive biases. Wang et al. (2022c) have an equivariant input path for symmetric state features, and a non-equivariant input path for non-symmetric state features. The non-equivariant path is used to generate dynamic filters (Jia et al., 2016) that obey equivariance constraints. Since equivariant networks are computationally more demanding than regular networks (Satorras et al., 2021; Kaba et al., 2023; Luo et al., 2024), we enforce symmetry in latent space without the need to build equivariant paths.

## 6 CONCLUSION

This paper presents an approach for incorporating prior knowledge of symmetry groups into abstract representations of MDPs. The method learns the weights of a state encoder and latent MDP dynamics to shape the abstract representations in a in a way that reflects the dynamics of the original environment. By enforcing geometric priors, the approach achieves better generalization compared to unstructured baselines. We show improved generalization on environments characterized by different combinations of symmetry groups and non-symmetry groups, including the high dimensional 3d first-person view environment VizDoom.

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

## A   LIMITATIONS

This work focuses on two fundamental symmetry groups—translations and rotations—which are prevalent in both representation learning and downstream reinforcement learning tasks. These choices provide a principled and tractable setting to demonstrate the benefits of incorporating geometric priors.

That said, our framework does not yet extend to more complex or task-specific symmetries (e.g., scaling, affine, or discrete groups). While technically feasible, such extensions would require additional development in defining group actions and enforcing structure in latent spaces. We leave this as a promising direction for future research.

## B   EXPERIMENTATION SET-UP

### B.1   ABSTRACT WORLD MODEL LEARNING ALGORITHM

---

**Learning world models with geometric priors**

Algorithm parameters: A discrete-time deterministic MDP $(\mathcal{S}, \mathcal{A}, \mathrm{T}, \mathrm{R}, \gamma)$.
Define the abstract state space $\mathcal{Z}$ and the group action operator $\oplus$.
Initialize an encoder $\varphi : \mathcal{S} \rightarrow \mathcal{Z}$, a transition model $\tau : \mathcal{Z} \times \mathcal{A} \rightarrow \mathcal{Z}$, a learning rate $\eta$, a replay buffer $\mathcal{D}$, and an exploration policy $\pi_{\mathrm{explore}}$.
Initialize the learnable parameters $\boldsymbol{\theta} := (\theta_{\mathrm{enc}}, \theta_{\mathrm{trans}})$.
**foreach** *episode* **do**
    Start at the initial state $s_0 \sim \mu_0$
    `// Phase 1: Collect empirical transitions from random`
    `    walking`
    **for** $t \leftarrow 0$ **to** $T - 1$ **do**
        Sample action $a_t \sim \pi_{\mathrm{explore}}(s_t)$.
        Observe the next state $s_{t+1} := \mathrm{T}(s_t, a_t)$.
        Append the tuple $(s_t, a_t, r_t, s_{t+1})$ to replay buffer $\mathcal{D}$.
    **end for**
    `// Phase 2: World model learning`
    **for** $i \leftarrow 0$ **to** $N - 1$ **do**
        Sample a batch of $M$ experience $\{(s_t, a_t, r_t, s_{t+1})_i\}_{i=1}^M$ uniformly from replay buffer $\mathcal{D}$.
        Obtain the latent states $\{(z_t, z_{t+1})\}_{i=1}^M$ from the encoder:
$$z_t^{(i)} := \varphi(s_t^{(i)}); \quad z_{t+1}^{(i)} := \varphi(s_{t+1}^{(i)})$$
        Obtain next abstract state predictions $\{(\hat{z}_{t+1})\}_{i=1}^M$ given the current abstract state and action:
$$\hat{z}_{t+1}^{(i)} := \tau(z_t^{(i)}, a_t^{(i)}) = z_t^{(i)} \oplus \Delta(z_t^{(i)}, a_t^{(i)})$$
        Compute the world model learning loss $\mathcal{L}_{\mathrm{abstract}}(\boldsymbol{\theta})$.
        Update the networks:
        $\theta_{\mathrm{enc}} \leftarrow \theta_{\mathrm{enc}} - \eta \nabla_{\theta_{\mathrm{enc}}} \mathcal{L}_{\mathrm{abstract}}(\boldsymbol{\theta})$
        $\theta_{\mathrm{trans}} \leftarrow \theta_{\mathrm{trans}} - \eta \nabla_{\theta_{\mathrm{trans}}} \mathcal{L}_{\mathrm{abstract}}(\boldsymbol{\theta})$
    **end for**
**end foreach**

---

**Algorithm 1:** *(Learning world models with geometric priors)* In practice, we learn the world model from the empirical transitions induced by a fixed policy $\pi_{\mathrm{explore}}$. Since world model learning is inherently an off-policy algorithm, our approach consists of two phases. First, for each episode, we collect experience tuples by interacting with the MDP. Then, we update the network parameters $\boldsymbol{\theta} := (\theta_{\mathrm{enc}}, \theta_{\mathrm{trans}})$ by jointly optimizing for the $\mathcal{L}_{\mathrm{abstract}}(\boldsymbol{\theta})$ calculated from the sampled mini-batches.

For most experiments, we parameterize the encoder $\varphi$ and the transition model $\tau$ with multi-layer perceptrons (MLPs) consisting of two hidden layers, each followed by `Tanh` activations. For the experiments on Vizdoom (Section 4.3), we use a convolutional neural network (CNN) to parameterize the encoder. We train both networks jointly for $50000$ gradient updates using the `RMSProp` optimizer with momentum. The batch size is fixed at 64 samples sampled uniformly from the replay buffer $\mathcal{D}$, and the learning rate is set between $10^{-5}$ and $10^{-4}$. We apply gradient clipping to stabilize training, with the clipping parameter set to 0.5.

## B.2 EXPERIMENTS DETAILS OF TORUS (SECTION 4.2) AND MINIGRID (SECTION 4.3)

In the Torus and MiniGrid environments, samples are collected using a random policy. No weighting factors were applied to balance the loss functions. All experiments for these two environments were conducted on a single Apple M3. Hyperparameters are listed in Table 2. The neural networks parameterizing the encoder $\varphi$ and the transition model $\tau$ are described in Table 3 and Table 4, respectively.

Table 2: Hyperparameters for Torus and MiniGrid.

| Parameter | Torus | MiniGrid |
|---|---|---|
| Learning rate | $10^{-4}$ | $10^{-4}$ |
| Batch size | 32 | 64 |
| Training steps | $50,000$ | $50,000$ |
| Gradient normalization | 0.5 | 0.5 |
| Number of valid transitions | 50 | 184 |

Table 3: Encoder architecture

| Layer | Layer Configuration |
|---|---|
| 1 | Dense (32 neurons, activation = `tanh`) |
| 2 | Dense (32 neurons, activation = `tanh`) |
| 3 | Dense (2 neurons (Torus) / 3 neurons (MiniGrid)) |

Table 4: Transition model architecture

| Layer | Layer Configuration |
|---|---|
| 1 | Dense (32 neurons, activation = `tanh`) |
| 2 | Dense (2 neurons (Torus) / 3 neurons (MiniGrid)) |

## B.3 EXPERIMENTAL DETAILS OF VIZDOOM (SECTION 4.3)

Our currently modification of VizDoom is based on the public implementation which can be found at https://github.com/Farama-Foundation/ViZDoom. A custom map (see Figure 9) was designed to evaluate our method in a high-dimensional setting. The map consists of a single room with textured walls, enabling the agent to localize itself within the environment. At each time step, the agent receives a reward of 0. For the experiment, images are downscaled to $64 \times 64$. All the experiments on VizDoom were conducted on a single Geforce RTX 3090. Hyperparameters are listed in Table 5. Moreover, the symmetrized version of infoCNE (Eysenbach et al., 2024) is used on the RL downstream task:

$$\mathcal{L}_{\text{InfoNCE}}(\theta_{\text{enc}}, \theta_{\text{trans}}) := \frac{1}{2}\mathbb{E}\left[-\log \frac{\exp\left(-d(\hat{z}_{t+1}, z_{t+1})/t\right)}{\exp\left(-d(\hat{z}_{t+1}, z_{t+1})/t\right) + \sum_{z_{t+1}^-} \exp\left(-d(\hat{z}_{t+1}, z_{t+1}^-)/t\right)}\right]$$

$$+ \frac{1}{2}\mathbb{E}\left[-\log \frac{\exp\left(-d(z_{t+1}, \hat{z}_{t+1})/t\right)}{\exp\left(-d(z_{t+1}, \hat{z}_{t+1})/t\right) + \sum_{\hat{z}_{t+1}^-} \exp\left(-d(z_{t+1}, \hat{z}_{t+1}^-)/t\right)}\right].$$

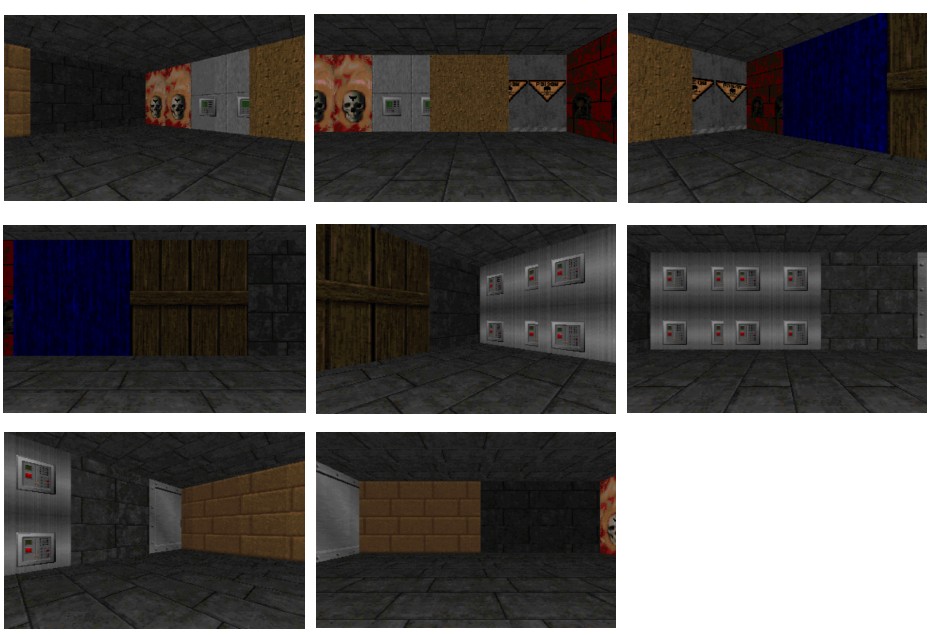

Figure 9: Custom map used in VizDoom experiment.

This variation of infoCNE improves stability in high-dimensional environments. The architectures of the encoder and transition model are detailed in Table 6 and Table 7, respectively.

Table 5: Hyperparameters for VizDoom.

| Parameter | VizDoom |
|---|---|
| Learning rate | $10^{-4}$ |
| Batch size | 200 |
| Training steps | $50,000$ |
| Gradient normalization | 0.5 |
| Dataset size | $100,000$ |

Table 6: Encoder architecture

| Layer | Layer Configuration |
|---|---|
| 1 | Conv2D (32 channels, $4 \times 4$ kernel, activation = tanh) |
| 2 | Conv2D (64 channels, $4 \times 4$ kernel, activation = tanh) |
| 3 | Conv2D (128 channels, $4 \times 4$ kernel, activation = tanh) |
| 4 | Conv2D (256 channels, $4 \times 4$ kernel, activation = tanh) |
| 5 | Dense (128 neurons, activation = tanh) |
| 6 | Dense (64 neurons, activation = tanh) |
| 7 | Dense (32 neurons, activation = tanh) |
| 8 | Dense (3 neurons) |

Table 7: Transition model architecture

| Step | Layer Configuration |
|------|---------------------|
| 1 | Dense (32 neurons, activation = `tanh`) |
| 2 | Dense (32 neurons, activation = `tanh`) |
| 3 | Dense (32 neurons, activation = `tanh`) |
| 4 | Dense (3 neurons) |

## C    METRICS USED

The Mean Reciprocal Rank (MRR) metric is defined as:

$$\text{MRR} = \frac{1}{N} \sum_{i=1}^{N} \frac{1}{\text{rank}_i}$$

where:

- $N$ is the total number of test instances (queries),
- $\text{rank}_i$ is the rank position of the correct data point in the ordered list of predictions for the $i$-th instance.

The hit at k (H@k) metric is defined as:

$$\text{H@k} = \frac{1}{N} \sum_{i=1}^{N} \mathbb{1}(\text{rank}_i \leq k)$$

where:

- $N$ is the total number of test instances (queries),
- $\text{rank}_i$ is the rank position of the correct data point for the $i$-th instance,
- $\mathbb{1}(\text{rank}_i \leq k)$ is an indicator function that equals 1 if the correct data point is ranked within the top $k$ positions, and 0 otherwise.

## D    GENERALIZATION TO UNSEEN TRANSITIONS

### D.1    MINIGRID

Figure 10 and Figure 11 illustrate the learned abstract representations of the world model with and without geometric priors. Our method produces a structured, simple, and highly predictable representation, leading to improved generalization across different data regimes (see Figure 12 and Figure 13).

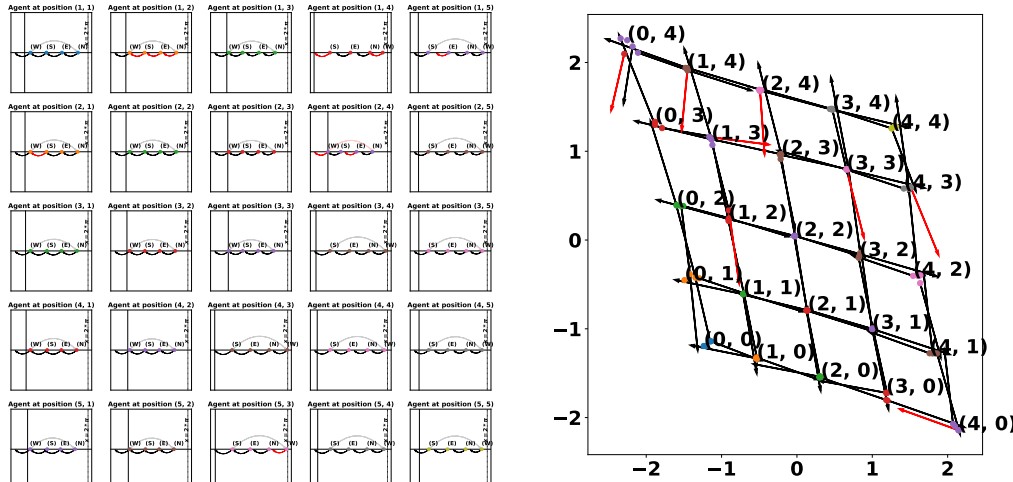

Figure 10: **Generalization to unseen transitions with geometric priors on MiniGrid** $5 \times 5$**.** The red arrows denote the unseen transitions during training (10% of the total valid transitions). As shown above, our method accurately predicts these transitions. (*Left*) Latent visualization of the first latent subspace $z^{(1)} \in \mathbb{R}/2\pi\mathbb{Z}$. (*Right*) Latent visualization of the second and third latent subspace $(z^{(2)}, z^{(3)}) \in \mathbb{R}^2$.

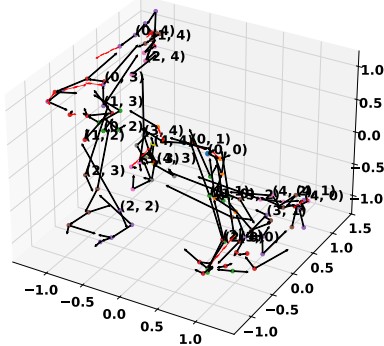

Figure 11: Abstract world model without geometric priors fails to generalize to unseen transitions in MiniGrid.

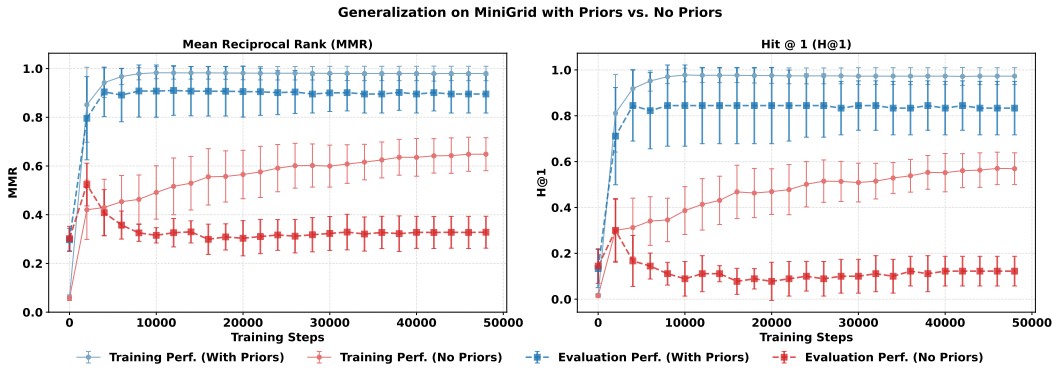

Figure 12: Generalization performance to unseen transitions (10% of the total valid transitions) on MiniGrid.

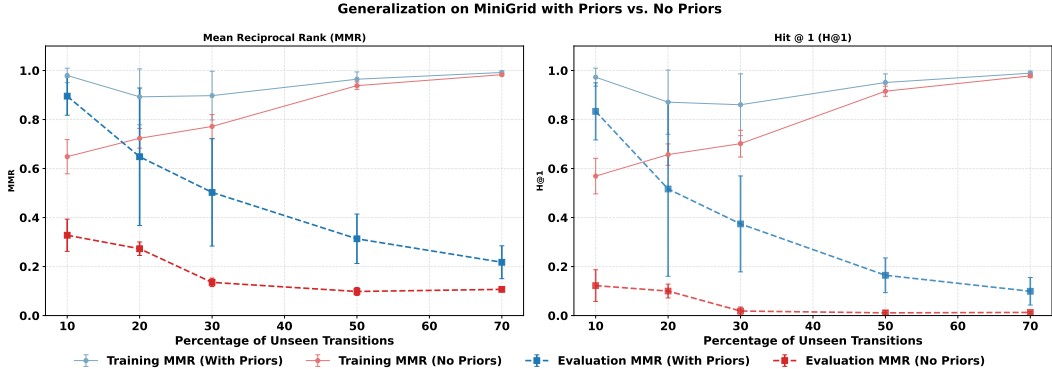

Figure 13: Generalization performance on MiniGrid with varying percentages of unseen transitions used as the validation set.

## D.2 TORUS

Similar to MiniGrid, we present quantitative results on generalization capabilities across different data regimes in Figure 14 and Figure 15.

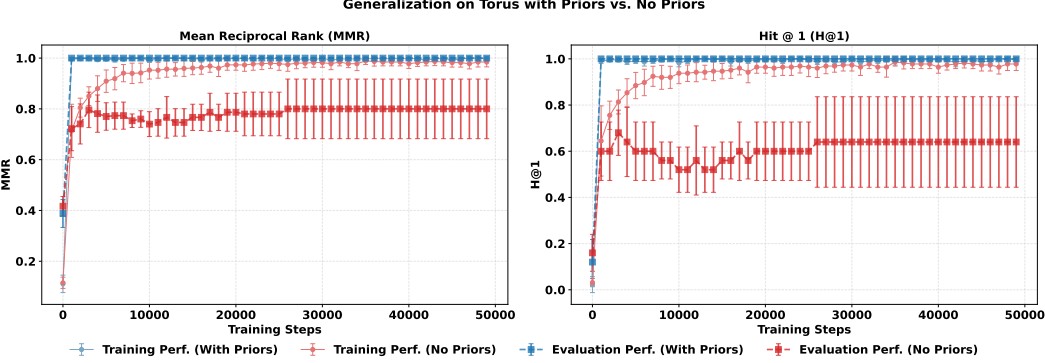

Figure 14: Generalization performance to unseen transitions (10% of the total valid transitions) on Torus.

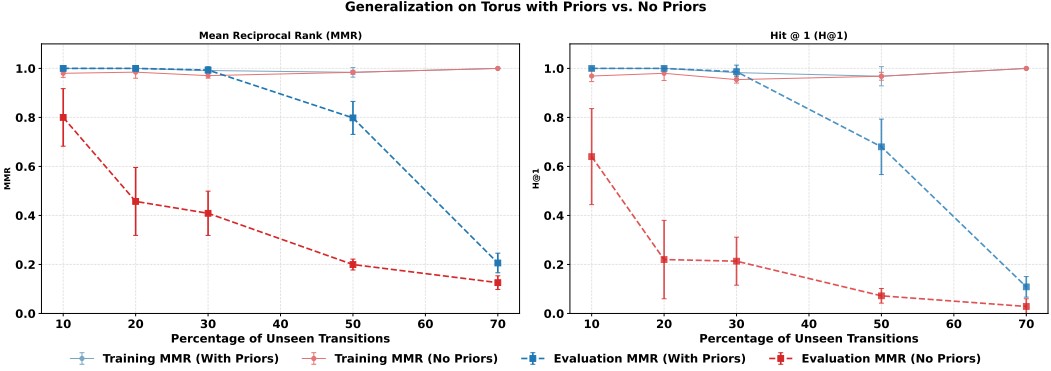

Figure 15: Generalization performance on Torus with varying percentages of unseen transitions used as the validation set.

### D.3 VIZDOOM

On VizDoom, our method outperforms abstract world models without prior knowledge when trained on 80% and 40% of the original dataset. Notably, the evaluation performance of the prior-informed agent exceeds even the training performance of the agent without priors (see Figures 16 and 17).

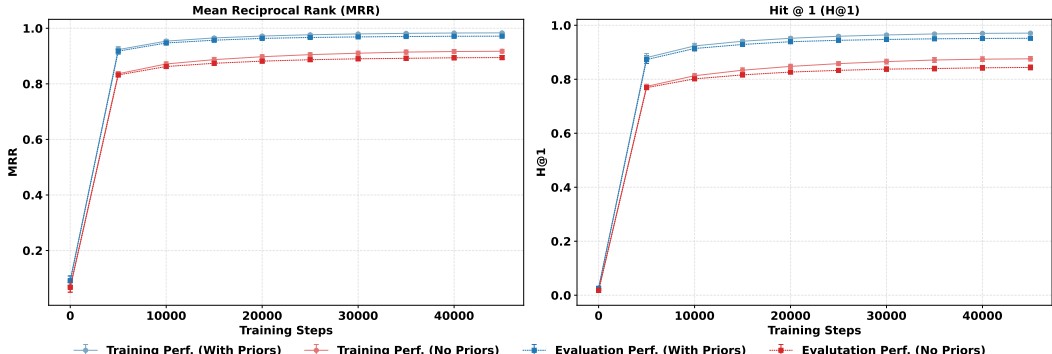

Figure 16: Vizdoom generalization with 80000 transitions (80% of the original dataset).

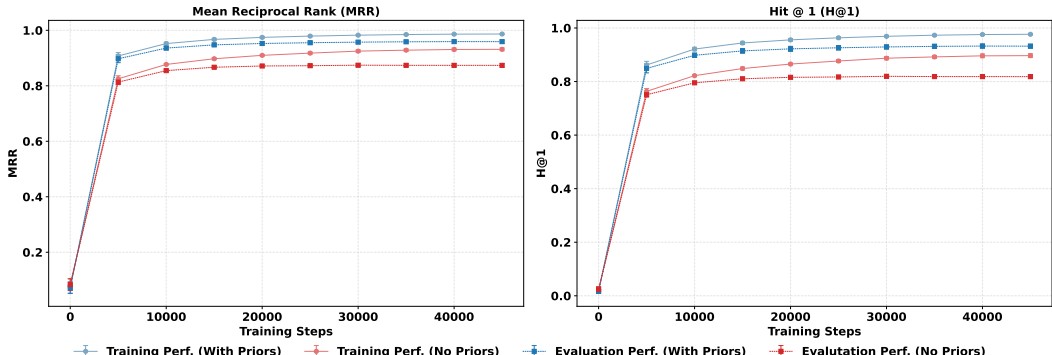

Figure 17: Vizdoom generalization with 40000 transitions (40% of the original dataset).

# E  ABLATION STUDY: EFFECT OF THE DISENTANGLEMENT LOSS

To evaluate the role of the disentanglement loss 11, we removed it on the VizDoom task and analyzed the impact on both performance and the structure of the abstract space. Surprisingly, performance on MMR and H@1 remained unchanged, and the latent space preserved a well-structured geometry(see Figure 18 and Figure 19). These findings suggest that the geometric constraints on the latent space alone are sufficient to guide the model. This observation opens the door to tackling more complex environments, where it is not known a priori which actions should influence which abstract features.

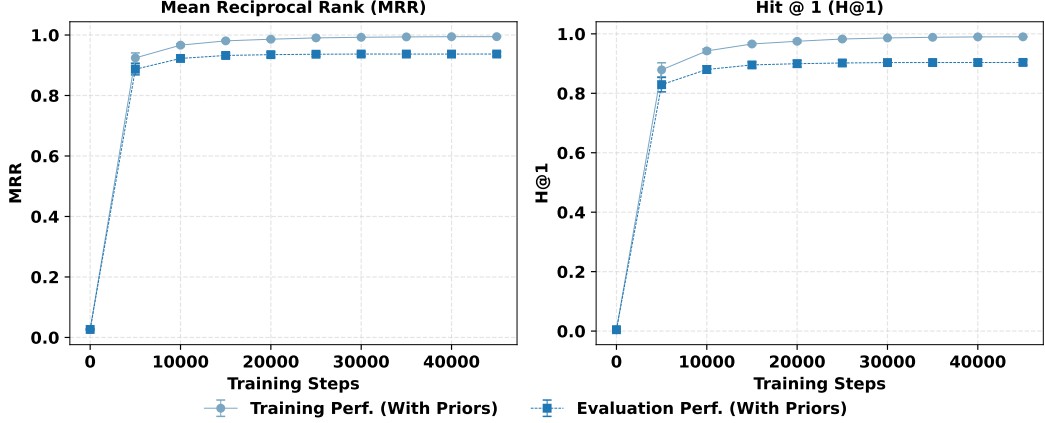

Figure 18: Vizdoom generalization with 10000 transitions (10% of the original dataset) without the loss 11

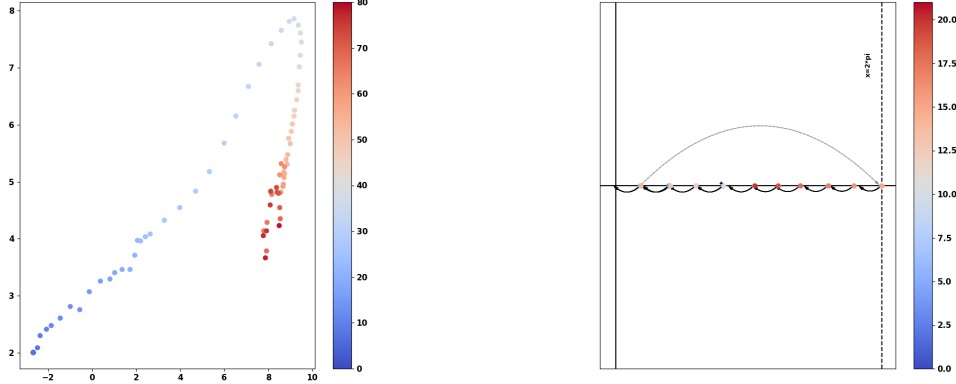

Figure 19: Latent space for the VizDoom environment withtout the loss 11. (*Left*) Latent subspace $(z^{(2)}, z^{(3)})$ that encodes spatial information. (*Right*) Latent subspace $z^{(1)}$ that encodes orientation.

# F  DOWNSTREAM REINFORCEMENT LEARNING TASKS

## F.1  EXPERIMENTATION SET-UP

The general task is for the agent to reach a designated location in the world. For every time step, it receives $-1$ as a reward and terminates the game as it reaches the goal. We propose a lightweight version of model-based augmentation by leveraging the learned abstract world model to generate synthetic one-step transitions for all actions at the abstract state level. This approach is both simple to implement and computationally efficient:

- No additional environment interaction is allowed—augmentation is performed solely on the replay buffer $\varphi$.
- For each stored transition $(s, a, r, s')$ in the dataset $\mathcal{D}$:
    1. Encode the state $s$ into its abstract representation $z$ using the frozen encoder.
    2. For all possible actions $a'$, use the learned transition model $\tau$ to predict the next abstract state $z'$ and the reward model to predict the reward $r'$.
    3. Add the synthetic tuple $(z, a', r', z')$ to the training batch $\mathcal{D}$ for Q-learning.
- It complements abstract-state Q-learning by densifying the data and improving generalization in the abstract space.
- It helps alleviate the *data sparsity problem* in offline reinforcement learning by enabling virtual exploration via the learned world model.

We evaluate three methods using a fixed offline dataset $\mathcal{D} = \{(s_i, a_i, r_i, s'_i)\}_{i=1}^{N}$ collected from environment interactions. All methods use Q-learning as the core algorithm for policy evaluation and improvement. The experiments proceed as follows:

1. **Pure Q-learning (Baseline)**
    - Sample mini-batches from $\mathcal{D}$.
    - Apply standard Q-learning updates using gradient descent to minimize the temporal-difference error:
$$\mathcal{L}_{\mathrm{Q}} := \mathbb{E}_{(s,a,r,s')\sim\mathcal{D}} \left[ \left( Q(s,a) - \left[ r + \gamma \max_{a'} Q(s', a') \right] \right)^2 \right].$$

2. **Q-learning combined with an Abstract World Model**
    - Train a world model consisting of a state encoder $\varphi(s) = z$, a transition model $\tau(z, a) \to z'$, and a reward model $\mathrm{r}(z, a) \to r$ using dataset $\mathcal{D}$.
    - Freeze the world model and proceed as follows:
        – For each $s$ in a tuple $(s, a, r, s') \in \mathcal{D}$, encode $s$ to $z = \varphi(s)$.
        – For all actions $a' \in \mathcal{A}$:
            * Predict $\hat{z}' = \tau(z, a')$ and $\hat{r} = \mathrm{r}(z, a')$.
            * Form synthetic tuples $(z, a', \hat{r}, \hat{z}')$ and add to the training set.
        – Apply Q-learning in the abstract space using both original and synthetic transitions.
    Two cases are considered:
    (2a) **World-model without geometric priors**: The abstract latent space is learned via a combination of $\mathcal{L}_{\mathrm{trans}}$ and $\mathcal{L}_{\mathrm{entropy}}$.
    (2b) **World-model with geometric priors**: The abstract latent space is equipped with algebraic structures as proposed.

On Torus and MiniGrid, the training set $D$ is $80\%$ of the total valid transitions of the underlying MDPs. The remaining $20\%$ are not accessible during the Q-learning step by any means. On VizDoom, a dataset of 150.000 transitions is collected from a random policy. We report the mean and standard deviation of cumulative rewards over multiple training seeds for all the variants.

F.2    VIZDOOM NAVIGATION TASK

In Section 4.5, the agent's goal is to reach a target position as quickly as possible (see Figure 20). At each time step, the agent receives a reward of $-\frac{||\text{agent position}-\text{target position}||_2}{\text{max distance}}$. If the normalized distance between the agent's position and the target position falls below $0.1$, the agent receives a reward of 10, and the episode terminates. Moreover, the episode ends after a maximum of 2500 steps.

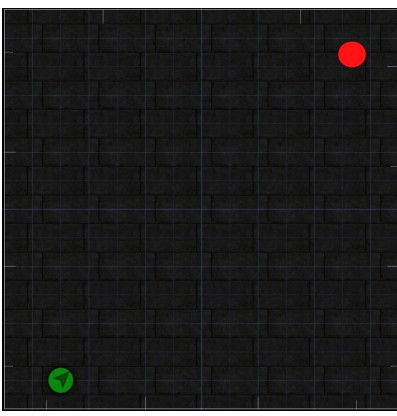

Figure 20: Downstream RL task in VizDoom. The agent starts at the green dot and aims to reach the red dot as quickly as possible.

