# OpenReview forum: "Learning Abstract World Models with a Group-Structured Latent Space"
_ICLR.cc/2026/Conference — Submitted to ICLR 2026_

### Official Review · Reviewer_Ney7 · 2025-10-27

**Soundness:** 2
**Presentation:** 2
**Contribution:** 1
**Rating:** 2
**Confidence:** 4

**Summary:**

The authors propose a method for training world models with geometric priors, capturing the latent group structure present in environments such as Torus-worlds and WizDoom. The authors construct geometric priors by making actions either be translations or modular addition, and use a loss function that encourages disentanglement between action effects and smaller latent state space sizes. The dynamics are learnt by minimizing an infoNCE loss. The authors show positive results for prediction and control in toy tasks and a pixel-based task.

**Strengths:**

* Systematic generalization in world models is important field of study
* Performance is reported in both prediction and RL settings.
* The paper is well motivated.

**Weaknesses:**

* My main concern lies on the construction of the priors. In the torus world it makes sense that the quotient group captures the structure of the environment, but in more complex environments its not so easy to say. In atari games and continuous control settings (like mujoco) it's not clear whether its feasible to use priors meaningfully.
* Related to this: the disentanglement loss seems to require that one pre-specifies the latent dimensions that each action should act on. While I get that the premise is that it's useful when one has priors, it puts significant constraints on the representation learning that might be necessary for complex tasks.
* Another thing I find puzzling is that each action only affects its own subspace. So each action acts on a different part of the environment. This seems very restrictive, often actions affect overlapping subspaces (e.g. moving left vs moving right).
* The environments the model is evaluated on are very simple.
* Many works deal with the question of learning robust world models that generalize systematically. Few of these are mentioned and none are tested against the model the authors propose.
* For instance, [1] learns the effect of actions as a linear transformation. [2] follows this up and makes dynamics softly state-invariant, giving actions consistent effects across the states of the environment. Lastly, [3] and [4] uses KL regularization and a target network respectively to make the latent dynamics model only contain information relevant for solving the task. These works need to be cited, as far as I see only [4] is cited.

Overall, I think the proposed model has some important flaws: It seems like one needs to specify the group action prior, the dimensions that the actions can affect, and assume that actions have orthogonal effects.

Second, the authors test the model on quite simple tasks. It would be interesting to see how this method fares in more complex environments, for instance mujoco/dm control suite.

Lastly, the authors should cite the relevant work mentioned above (citation [1], [2] and [3]).

If these points are addressed properly I'm happy to increase my score.

References:

[1] Watter et al. "Embed to control: A locally linear latent dynamics model for control from raw images." NeurIPS 2015

[2] Saanum et al. "Simplifying latent dynamics with softly state-invariant world models." NeurIPS 2024

[3] Zhu et al. "Repo: Resilient model-based reinforcement learning by regularizing posterior predictability." NeurIPS 2023

[4] Hansen et al. "Td-mpc2: Scalable, robust world models for continuous control." ICLR 2023

**Questions:**

* How important is the choice of prior? What happens if you train the model on the Torus parameterizing the dynamics using translation (instead of modular arithmetic)
* Can this method scale up to more challenging environments with continuous actions? If so, how is the prior determined and how are the dimensions that each action affects determined?
* What is the purpose of the volume penalty - does this actually improve
* Quite puzzling that the policy with the AWM performs so poorly. This is simply a world model with a latent space learned with a contrastive loss. Similar methods have been proposed and have been shown to work well (RePo [citation]). Why does this model fail in this setting?

---

### Official Review · Reviewer_fE6i · 2025-10-30

**Soundness:** 3
**Presentation:** 3
**Contribution:** 1
**Rating:** 2
**Confidence:** 4

**Summary:**

This paper proposes incorporating translation/rotation priors into world models to improve both their predictive power and the performance of downstream reinforcement learning tasks. These priors are expressed by group theory, effectively encoding environmental symmetries of translation and rotation by constraining the latent dynamics model. The world model is trained solely using contrastive learning, abandoning traditional reconstruction objectives, possibly because reconstruction becomes impossible with such a low-dimensional latent space.
The authors claim that applying translational and rotational priors in navigation tasks improves the model's ability generalize across locations and angles.

**Strengths:**

- strong theoretical foundations; illustrated with intuitive examples
- compelling concept of efficient, low dimensional representations suitable for downstream tasks
- paper is well structured

**Weaknesses:**

- the approach requires prior knowledge about the symmetries in the environments. while the authors state that using more complex symmetries is up to future research, symmetries have to be discovered by humans beforehand, which limits generic applicability
- the function σ  is introduced to disentangle the representation, while details how this is implemented are missing. Moreover, the results in Appendix E indicate that this additional component may actually not be necessary.

- It is unclear to what extent the transition model is influenced by the observations. For the Passage and Torus environment, the observation already contains position information, using the translation / rotation priors would imply only learning a mapping from one-hot encoded information to actual positions/rotations. For the VizDoom environment, it seems like the authors used a fixed initial state. In the absence of noise the future latent state (orientation + position) could be predicted without actually using observations at all by simply using knowledge about the initial position and the performed action sequence. it seems like the prior itself could already solve the problem.

- Generalization to more complex environments unclear.

- the comparison with a low-dimensional (2-3 dimensions) world model without the proposed priors seems unfair. The comparison model is solely trained with contrastive learning and reward prediction in mind while contraint to an extreme bottleneck in the latent space.

**Questions:**

- The role of σ  is not clear and missing some explanations. How is it derived? Is it pre-defined or learnt?
- Are the (potentially high-dimensional) observations actually used for learning the latent representation (especially for the VizDoom environment) or is the prior and initial state sufficient to predict the future? can you run additional experiments with randomized intitial position?
- Figure 6: is the plot showing the position inferred from high dimensional observations or is the plot generated by starting in the initial state and unrolling the world model?
- Is the learnt encoder able to infer the correct latent state (i.e. orientation and position) from the high dimensional observation alone?
- If the symmetries are known beforehand, why even use model bases RL? The idea appears to be more closely related to SLAM approaches.

---

### Official Review · Reviewer_SyM3 · 2025-10-31

**Soundness:** 3
**Presentation:** 2
**Contribution:** 2
**Rating:** 4
**Confidence:** 3

**Summary:**

This paper proposes a world-modeling framework that improves interpretability and sample efficiency by giving the latent state space a geometric structure and defining transitions with a fixed group operation. In practice, the next latent state is computed by applying a human-specified symmetry operation (for example, modular addition on a circle or torus) to the current latent, using a learned, action-conditioned step. This makes the latent dynamics equivariant-by-construction to the chosen prior and induces an abstract MDP over latents whose transitions respect that symmetry while still allowing the network to learn the step size and direction.

**Strengths:**

- The work tackles a highly relevant problem of improving world model latent and dynamical interpretability.

- While some earlier work has investigated learning latent representations of high-dimensional state/observation spaces by explicitly specifying functions to which the representations should be invariant/equivariant, this paper takes the solution further by selecting a structured latent space where a set of actions produces "legal" transitions, according to some observed geometric behaviors.

- The results demonstrate the empirical value of encoding symmetries in world models.

**Weaknesses:**

- Your MDP definition is missing the stochasticity of the transition kernel T. Specifically, T should be a mapping T: S x A x S -> [0, 1]. It is okay that you only consider the deterministic case T: S x A x S -> {0, 1} throughout the rest of the formulation and solution, but the definition should align with the mainstream understanding of an MDP. What you have is more of a state-and-reward state machine.

- By replacing the learned transition's composition rule with a fixed, equivariant operator, tied to some human-derived geometric prior, the model will be unable to express valid transitions that fall outside this operator family. While this is fine for simple free-roam experiments like the gridworld robot, it is hard to see how this would work when there are unexpected discontinuities in the scene (e.g., placing an obstacle in the gridworld). DNNs often help precisely because they discover patterns we cannot pre-enumerate; here, the hypothesis class is constrained to transitions that factor through the chosen operator.

- The paper misses closely related prior work that learns world model latents with explicit invariances/equivariances:
	- Zhang et al. Invariant Causal Prediction for Block MDPs. Proceedings of the 37th International Conference on Machine Learning, PMLR 119:11214-11224, 2020. https://proceedings.mlr.press/v119/zhang20t.html
	- Zhang et al. Learning Invariant Representations for Reinforcement Learning without Reconstruction. Proceedings of the 37th International Conference on Machine Learning, 2021. https://openreview.net/forum?id=-2FCwDKRREu
	- Peper et al. Four Principles for Physically Interpretable World Models. Proceedings of the International Conference on Neuro-symbolic Systems, PMLR 288:66-89, 2025. https://proceedings.mlr.press/v288/peper25a.html

**Questions:**

1. As mentioned earlier, this abstract world model would suffer from dynamics with discontinuities. Can we find other priors that handle these cases (e.g., a prior based in elastic collision for scenarios with rigid obstacles)?

2. Virtually all world models are abstract in the sense that they are not explicitly formulated in terms of the sensor data. Typically, this property is referred to as "latent". Do you call all (not just your) world models "abstract" in reference to some other abstraction property? If not, it is best to align with the standard vocabulary of the field.

3. How does your work relate to the three references provided above?

---

### Meta-Review · Area_Chair_63f8 · 2025-12-26

**Summary:**

This paper learns latent world models, using an InfoNCE loss, but with additional symmetric structure about the transitions in the latent space. The idea of adding structure to the latent space dynamics is intellectually useful. However, there were concerns about the practicality of the method and the strength of the evaluation.

**Reviewer Concerns:**

Most concerns were about the initial prior knowledge assumed (about the symetry), the simplicity of the test environments, and on comparisons with related work.
I agree with the reviewers - these concerns must be addressed before this work can be published.

The authors did not respond with a rebuttal.

**Reviewer Scores:**

4,2,2

---

### Decision · Program_Chairs · 2026-01-26

Reject